# Time-Series Representation Learning in Topology Prediction for Passive Optical Network of Telecom Operators

**DOI:** 10.3390/s23063345

**Published:** 2023-03-22

**Authors:** Haoran Zhao, Yuchen Fang, Yuxiang Zhao, Zheng Tian, Weinan Zhang, Xidong Feng, Li Yu, Wei Li, Hulei Fan, Tiema Mu

**Affiliations:** 1Department of Computer Science and Engineering, Shanghai Jiao Tong University, Shanghai 200240, China; zhao.hr@sjtu.edu.cn (H.Z.);; 2China Mobile (Zhejiang) Innovation Research Co., Ltd., Hangzhou 310016, China; 3School of Creativity and Art, ShanghaiTech University, Shanghai 201210, China; 4Digital Brain Laboratory, Shanghai 200072, China; 5Computer Science Department, University College London, London WC1E 6BT, UK

**Keywords:** topology prediction, passive optical network, noise reduction

## Abstract

The passive optical network (PON) is widely used in optical fiber communication thanks to its low cost and low resource consumption. However, the passiveness brings about a critical problem that it requires manual work to identify the topology structure, which is costly and prone to bringing noise to the topology logs. In this paper, we provide a base solution firstly introducing neural networks for such problems, and based on that solution we propose a complete methodology (PT-Predictor) for predicting PON topology through representation learning on its optical power data. Specifically, we design useful model ensembles (GCE-Scorer) to extract the features of optical power with noise-tolerant training techniques integrated. We further implement a data-based aggregation algorithm (MaxMeanVoter) and a novel Transformer-based voter (TransVoter) to predict the topology. Compared with previous model-free methods, PT-Predictor is able to improve prediction accuracy by 23.1% in scenarios where data provided by telecom operators is sufficient, and by 14.8% in scenarios where data is temporarily insufficient. Besides, we identify a class of scenarios where PON topology does not follow a strict tree structure, and thus topology prediction cannot be effectively performed by relying on optical power data alone, which will be studied in our future work.

## 1. Introduction

Passive optical network (PON) does not contain any active electronic devices or power source in the optical distribution network (ODN), but is composed of passive devices such as optical splitters [1], as is shown in Figure 1. On the one hand, the network has many advantages such as low cost and low resource consumption, thus widely used in fiber optical communication. On the other hand, the passiveness prevents each secondary splitter from sending information to the optical line terminal (OLT) about its higher and lower levels in the topology, which makes the tree structure of ODN unclear to the OLT. It is thus difficult to know which secondary splitter an optical network unit (ONU) belongs to.

Obtaining complete and accurate topology of PON is very important in practice. The management of PON consists of two parts, namely resource management and fault management. Resource management requires knowledge of the number of ONUs on each secondary splitter to understand the resource occupancy and determine the feasibility of capacity expansion. Fault management requires quick location of the secondary splitter over an ONU when it fails. Both of them require a clear structure of PON as a prerequisite. Therefore, methods to predict PON topology are in great demand.

In this paper, we fully explore the structure of PON as well as the features of data and propose a neural-network-based PON topology prediction solution (PT-Predictor) to identify which secondary splitter each ONU belongs to. The main contributions are concluded as follows:We firstly introduce neural networks into such problems as PON topology prediction and provide a base solution for these tasks, including a preprocessor, a scorer and a voter, which outperforms previous model-free methods.We design a useful GRU-CNN-ensemble-based scoring model (GCE-Scorer) to extract the relationship between ONUs and give their similarity scores, which achieves better performance over the base solution.We implement a max-average-aggregation algorithm (MaxMeanVoter) based on data analysis and a novel Transformer-based voter (TransVoter) to predict the secondary splitter for each ONU, using scores provided by the scorer.

We conduct experiments on an industrial dataset provided by telecom operators to show the effectiveness of PT-Predictor. And we further explore the contribution of each ingredient including GCE-Scorer and TransVoter. Besides, we combine data analysis and field study to identify the application scope of PT-Predictor. Within the scope, PT-Predictor is more effective and cost-efficient than previous manual and hardware methods, and serves as the best software-based method until now.

## 2. Problem Definition

One important fact of PON is that the information of belonging secondary splitter of each ONU is usually registered in the resource management systems (RMS) upon installation. However, under normal circumstances the accuracy is not guaranteed due to poor management. In some communities, the accuracy can be as low as 50% for the following reasons: (1) installation error: workers only make sure the ONUs function noramlly and do not assure the consistency between registered topology and actual installation. (2) removal: when an ONU is removed, its wires to the secondary splitter may not be removed. (3) migration: when an ONU is migrated from one place to another, its information in RMS is not necessarily updated. Therefore, the registered information has many errors, which we will refer to as noise of the ONU label hereinafter.

Nowadays, manual-based and hardware-based methods are most widely used for PON noise reduction, but there are shortcomings. Manual-based method requires telecom operators to hire third-party professional workers to check the ONUs one by one on site, which is normally expensive and time consuming and the quality of the work is not guaranteed. Besides, new installations and removals happen every year, resulting in changes in the topology structure and repetitive manual work. Hence, this approach can hardly be a reliable and sustainable solution. Hardware-based method requires reflective chips installed on splitters, which is costly and difficult to maintain.

For the sake of low cost and sustainability, software-based method becomes the main object of study. It utilizes time-series data from each ONU, e.g., the transmitted and received power, to conduct noise reduction. The physical mechanism is that when a secondary splitter is perturbed by external environments (e.g., wind, rain, or animals passing by), the optical power waveforms of the ONUs connected to that secondary splitter reveal similar features, as is shown in Figure 2a. By capturing these consistent fluctuations, it is possible to determine that these ONUs belong to the same secondary splitter. Then one can cluster them and tell whether their labels are errors.

However, this software-based method faces some challenges: (1) the signal acquisition is not continuous, thus not all the perturbations are caught; (2) only perturbations strong enough can be detected, and there are not many of them; (3) ONUs under the same secondary splitter do not necessarily show similar features depending on the location where perturbation happens. It can be seen in Figure 2b that some of the ONUs show a high degree of similarity while the others don’t. Due to the above challenges, current software-based methods such as directly clustering ONUs according to their optical power don’t work well, and more powerful methods are needed.

Given the background, we can formulate our problem in a more general form as follows: RMS provides a tree, as is shown in Figure 1, but the registered tree structure may be inconsistent with the actual situation. Leaves’ labels (their parents) can be wrong (noises), and need to be corrected based on the fact that leaves from the same parent show similar characteristics. In this paper, we develop better models to capture the similarity of leaves, which will be achieved by GCE-Scorer, and design better algorithms to perform noise reduction on the leaves’ labels, which will be completed by MaxMeanVoter and TransVoter. The overall methodology PT-Predictor combines these techniques to eventually help us obtain the correct tree structure.

## 3. Related Work

Previous methods have been concluded in detail for PON failure management [2,3] which is one important function of topology prediction, but these works are too much domain oriented. Our work focuses on topology prediction and can be extended to a more generic problem as explained in Section 2, so we pay more attention to generic techniques.

The key object studied in this paper is optical power, typical time-series data, so recurrent neural networks are usually applied, such as LSTM [4] and GRU [5]. Convolutional neural networks have also been improved to capture relationship over a longer time span. In field of audio generation, the dilated convolution used in WaveNet [6] achieves satisfactory performance. The exponential dilated causal convolution inspired by WaveNet is also used for universal time-series data [7]. These works show that both convolutional neural networks and recurrent neural networks function well in time series data, which is consistent with our work. Besides CNNs and RNNs, Zerveas et al. provide a general approach to apply Transformer [8] to time series prediction [9], which shows the performance of attention mechanism. Furthermore, a lot of improvements have been made to Transformer based on the features of time-series data [10]. For positional encoding, Lim et al. employs an LSTM to encode the location information [11], and Informer [12] encodes the timestamp and incorporates this information into the temporal data through a learnable model. For the architecture, Informer [12] applies a maximum pooling layer between attention blocks to perform downsampling, and Pyraformer [13] designs C-ary trees to capture fine-grained and coarse-grained features respectively. We learn from the above techniques and then design our NN-based model GCE-Scorer and TransVoter.

Most classification problems lie in supervised learning. Also, Franceschi et al. applies triplet loss to unsupervised learning through specially designed sampling method [7]. But our scenario is different. Our dataset provides complete topological information (labels), but a significant percentage of these labels is wrong. Therefore, the focus of our training process is to reduce the impact of these noisy labels. Peer loss uses peer prediction to reduce noise [14], which is based on the assumption that noise labels are uniformly distributed over the training dataset. By subtracting the loss of random samples simulating noise from that of normal samples, the impact of noise on training can be effectively reduced. Noise pruned curriculum loss uses curriculum loss to reduce noise [15]. The key of this method is to select training samples and use only the best ones for training. By eliminating training samples that have a higher probability of being noisy, noise pruned curriculum loss effectively avoids the participation of noisy data in training. In addition, Normalized Loss [16] and other methods can also reduce the impact of noise. Our work utilizes some of the methods and achieves good results.

## 4. Methodology

### 4.1. Base Solution

The structure of base solution is shown in Figure 3, including a preprocessor, a scorer and a voter. Optical power and topology data are firstly combined and preprocessed to get sequences, which will be suitable for neural networks to deal with. Various feature engineerings are conducted as is described in Section 4.2. These sequences are then sampled to get pairs, and the scoring model captures their similarity to give scores, which will be discussed in Section 4.3. Finally, these scores are aggregated by the voter to determine whether a label is wrong, as is shown in Section 4.5. Then we complete noise reduction and get the accurate topology.

The base solution already outperforms previous model-free methods. But we further explore different stages of the process and design GCE-Scorer, MaxMeanVoter and TransVoter to achieve better performance. The best methods in each stage make up the final methodology PT-Predictor.

### 4.2. Preprocessing and Feature Engineering

In preprocessing and feature engineering stage, we transform the raw data into sequences that can be easily processed by machine learning models and extract features from the raw data sufficiently. The overall process is roughly illustrated in Figure 4.

**Cutting.** For some ONUs, the span of time reaches half a year, and as is shown in Figure 2, the duration of the perturbation is mostly at the hourly level, so it is necessary to cut the optical power data over a long time into numerous shorter sequences.

**Filtering.** Ideally, the probe located on the ONU would perform data acquisition every 10 min, which is determined by the probe’s physical design. But in actual production, it is not always working, and there exists a significant percentage of cases where the probe returns null values. If the percentage of null values in a sequence is too high, meaningful perturbations may be skipped and messages such as null values are captured by the model, so sequences whose null rates are higher than 20% will be filtered out, which will be explained in Appendix B in detail.

**Normalization.** In actual production, there are various types of waveforms depending on communities, such as standard community and tenant community, and our work aims to build a unified model for all communities. Through experiments, we find that the mean and variance of optical power from these communities varies a lot, so we firstly normalize at the level of primary splitters.

In addition, the feature of perturbation needs to be captured as a priority in the model, but such basic information as mean and variance of different secondary splitters may also differ significantly. On the one hand, these differences affect the learning focus of the model and make it unable to capture the perturbation; on the other hand, the differences itself can play a good role in distinguishing the secondary splitters for some specific communities. Therefore, we experiment on different levels of normalization in Section 5.2, and select the best one for PT-Predictor.

**Linear Interpolation and Smooth Average.** After filtering, there are still a certain proportion of null values in the remaining sequences, so we fill the sequences using linear interpolation. Besides, as is shown in Figure 2, the waveform features include macroscopic scale (overall waveform trend) and microscopic scale (frequent up and down fluctuations). We use average smoothing to explore the impact of fluctuations at different scales, and select the best window size as 10 in our experiments.

### 4.3. Sampling and Scoring

In preprocessing stage, we transform raw data into sequences, and now we need to select the ones located at the same time and extract the features of waveform consistency. To achieve this, we design sequence-wise scoring models to give scores of the similarity for each sequence pair.

The key of scoring model is to process two sequences and get the scores. We design the model based on different units such as CNN and RNN. And we adopt various processing techniques, such as dealing with the difference of two sequences or separately processing two sequences. The experiments in Section 5.2 prove that our carefully designed GCE-Scorer achieves the best performance, which is an ensemble of a CNN-based model and a GRU-based model, as is shown in Figure 5.

In this subsection, we tentatively consider our work as a supervised case, i.e., the secondary splitter to which each ONU belongs is known (in fact this topology information has been provided by RMS, only that there is noise). Then we can select pairs of sequences at the same time and from the same secondary splitter. These pairs are labeled as positive, while the others as negative. The optimization is to enlarge the scores of positive pairs and reduce the scores of negative pairs.

### 4.4. Noise-Tolerant Techniques for Training

We have built a supervised sampling and scoring framework in Section 4.3, but it is based on the premise that the topology information provided by RMS must be accurate. In fact, the topology has noises.

Assume that there are *k* secondary splitters containing *n* ONUs under the specific primary splitter, and the noise rate of ONU’s label is α, i.e., each ONU has a probability α of being randomly labeled to other secondary splitters with equal chance. We firstly choose one target sequence. The topology information provided by RMS shows that the number of ONUs under the secondary splitter of the target sequence is m(m<n), and all ONUs have sequences sampled from the same time as the target sequence. Then, we sample another sequence and label the pair according to the information given by RMS. Denote the probability that the selected positive pair is truly negative as p(neg|pos) and the probability that the selected negative pair is truly positive as p(pos|neg). We have:(1)$$p\left(neg|pos\right)=\frac{\alpha (n-m-n\alpha )}{\left(k-k\alpha -1)(m-1\right)}$$
(2)$$p\left(pos|neg\right)=\frac{\alpha (km-m-n\alpha )}{\left(k-k\alpha -1)(n-m\right)}$$

The detailed calculation procedure is shown in Appendix A. We substitute n=24,

k=4,m=6,α=0.2 and then get:(3)p(neg|pos)=0.24p(pos|neg)=0.067

To be honest, the above calculation is based on some assumptions, and the actual situation is more complex. But these results are sufficient to show that the presence of noise has an impact on the sampling of positive and negative samples, especially positive samples. So, we explore methods to avoid the interference of noise during the training process.

#### 4.4.1. Peer Loss

The peer loss function reduces the effect of noise with the help of peer prediction, the mathematical principles of which are detailed in previous work [14] and the formula is as follows:(4)$$\ell_{peer}\left(f\left(x_{i}\right),y_{i}\right)=\ell \left(f\left(x_{i}\right),y_{i}\right)-\ell \left(f\left(x_{i_{1}}\right),y_{i_{2}}\right)$$

$\ell (f\left(x_{i}\right),y_{i})$ is the base loss function for training sample $\left\{x_{i},y_{i}\right\}$. $\left\{x_{i_{1}},y_{i_{2}}\right\}$ is sampled randomly from trainsets, and $\ell (f\left(x_{i_{1}}\right),y_{i_{2}})$ simulates the loss under pure noise and then is subtracted from the base loss function.

#### 4.4.2. Noise Pruned Curriculum Loss

The work of curriculum loss focuses on selecting better samples for training. For tasks with noisy labels, the better training samples are naturally those without noise, based on which noise pruned curriculum Loss (NPCL) is proposed, which is illustrated in previous work [15] in detail. The formula is as follows:(5)$$\begin{array}{c} L_{npcl}\left(x,y\right)=\min_{v\in {\left\{0,1\right\}}^{(1-\epsilon )n}}\max\left(\sum_{i=1}^{(1-\epsilon )n}v_{i}\ell \left(x_{i},y_{i}\right),\left(1-\epsilon \right)n-\sum_{i=1}^{(1-\epsilon )n}v_{i}\right) \end{array}$$

$\ell (x_{i},y_{i})$ is the base loss function for traning sample $\left\{x_{i},y_{i}\right\}$. *n* is the number of all training samples and ϵ is the expected noisy rate. NCPL uses vector v to filter out the samples with higher loss, corresponding to highly similar negative pairs and quite different positive pairs, which tends to be noisy.

### 4.5. Data-Based Voting

In sampling and scoring stage, scoring model is able to give similarity scores of two sequences. The voter aggregates these scores to predict the secondary splitter for each ONU, which consists of two steps: getting the score of ONU pair from sequence pairs, and getting the score of splitter from ONU pairs. We analyze data features, and propose MaxMeanVoter, which applies different aggregation methods in the two steps.

In the first step, we notice that perturbation is a random event. Only a few positive pairs of sequences are able to capture the perturbation, as is shown in Figure 6, and the similarity of these pairs will be higher. So, we conduct max voting and the experiments in Section 5.2 prove our choice to be right.

For the second step, the principle is shown in Figure 7. For each ONU, its true label exists objectively, but is unknown to us. We can only obtain its noisy label. We first select a target ONU, and prepare a large number of other candidate ONUs. These candidate ONUs are divided into clusters (representing different secondary splitters) according to their noisy labels. When the noise rate is less than 50%, it is roughly guaranteed that the cluster with noisy label of *x* has the majority of candidate ONUs with true label of *x*. Therefore, the average similarity between the target ONU and the cluster with the same true label will be higher, then we get the right secondary splitter.

### 4.6. Model-Based Voting

MaxMeanVoter is designed considering data features. But simple maximum and average operations may still be not expressive enough to capture all the features of the scores, so we propose TransVoter, a self-attention based model to achieve higher expressiveness, whose principle is shown in Figure 8.

All the information of sequence pairs concerning one target ONU is a tensor of size a×d×b, where *b* denotes all the secondary splitters in the primary splitter, *d* denotes all the ONUs in each secondary splitter, and *a* denotes all the sequences between the target ONU and each candidate ONU, from the start collection time to the end time. The dimension of *a* and *b* are first integrated to get *g* vectors of size *d*. These vectors are then processed by a self-attention model to get the probability for each secondary splitter. Experiments in Section 5.2 prove TransVoter to perform better than simple data-based voter.

## 5. Experiments

### 5.1. Datasets

Datasets in this paper are provided by telecom operators. As of the completion of the experiments covered in this paper, they have provided us with the topology of 65,043 unchecked ONUs (the labels provided contain noise) and 3084 checked ONUs (the labels prove to be accurate), as well as the corresponding optical power. In our experiments, we use the noisy ONUs for training and the noiseless for testing. In the testing process, we first add noise to the noiseless ONUs at a random rate to obtain a noisy version, and then use PT-Predictor to conduct noise reduction and compare it with the noiseless version to judge the effect.

After the data preprocessing in Section 4.2, we choose the communities that still retain a large number of sequences (>200) as our test dataset, and they cover the OLTs shown in Table 1. Among them, data collection of OLT 0 and OLT 1 has reached 29 weeks, which is more reliable, and therefore serve as the representative communities for our experiments. Their ground truth topology is shown in Figure 9. The other communities have 2∼4 weeks of data collection and are slightly less reliable, but can still be analyzed similarly.

### 5.2. Results and Discussion

We conduct experiments on OLTs with different collection time and the results are shown in Table 2. It is not difficult to find the anomaly of OLT 1, about which we will analyze in detail in Section 5.3. Now we focus on the other OLTs. In Table 2, k% coverage means the top k% ONUs with the most confidence. Previous method refers to using FastDTW to calculate distances between ONUs after normalization and then clustering, which is a common method for time-series data before neural networks are introduced. In the base solution, we normalize the data on the level of OLT, use GRU as scoring model, and apply only average aggregation in the voting stage, which are all simple operations. And PT-Predictor adopts all techniques to achieve the best results.

We conduct further experiments on the influence of various techniques and the result is shown in Figure 10. We try different levels of normalization and Table 3 shows that normalization on sequences achieves the highest performance, which is consistent with our assumption in Section 4.2. We test various scoring models as well as FastDTW and Table 4 proves GCE-Scorer to be the most useful. In the two steps of data-based voting, we use different combinations, and Table 5 reveals that MaxMeanVoter performs the best, which corresponds to analysis in Section 4.5.

The datasets for experiments of data-based voter are different. We firstly go through the first two stages: preprocessing and scoring, with GCE-Scorer and obtain the scores for each sequences pairs. We divide these pairs into training dataset and testing dataset and then conduct voting experiments. The results are shown in Table 6. And TransVoter provides another leap of performance.

Although lots of improvements have been achieved, we have to notice that PT-Predictor performs obviously worse on other OLTs than OLT 0, and the collection time is worth paying attention to. To confirm this assumption, we conduct further experiments. We test how the accuracy grows as collection time increases and the results are in Table 7. The experiments show that as the amount of data increases, which indicates the growth of effective perturbations captured, the accuracy is improved. The improvements can even be seen in OLTs with only 4 weeks of data collection, so higher performance is promising. Data provided is not sufficient by the time of completing this paper, but we will continue this study. And we will conduct more experiments in our future work with more follow-up data given in those temporarily insufficient datasets.

### 5.3. Anomaly Research and Analysis

Although 29 weeks of data are provided on OLT 1, the result is still poor, so we analyze the waveform of OLT 1 specifically.

The waveforms of two secondary splitters from OLT 1 are shown in Figure 11. It can be seen that ONUs under the same secondary splitter do not show similar waveforms, while ONUs under different secondary splitters show similar waveforms. This phenomenon is against the physical mechanism of PT-Predictor. For this reason, we specifically conduct a field study, and learn that in scenarios like OLT 1, ONU wires under different secondary splitters come together, so the optical network topology does not satisfy a strict tree structure. When perturbations occur at these sites, it is impossible to tell whether the similarity of waveforms is due to being under the same secondary splitter or twisted wires, with optical power data alone provided.

## 6. Conclusions

In this paper, we study the work of topology prediction for PON of telecom operators and provide a base solution for such problems in practice. Based on the base solution, we propose effective deep neural network ensembles with various noise-tolerant techniques integrated, and design novel data-based and model-based voters to make up the final methodology PT-Predictor. PT-Predictor is significantly better than previous manual, hardware, and software solutions in terms of accuracy and cost. In scenarios where data is sufficient, the prediction accuracy can be improved by 23.1% compared to model-free methods, and in scenarios where data is temporarily insufficient, the accuracy can also be improved by 14.8%. Besides, the accuracy grows obviously as data provided increases, so further results are promising and we will continue the experiments when more weeks of optical power are collected.

Also, we explore the application scope of PT-Predictor and identity a class of scenarios where PT-Predictor doesn’t work well. Through data analysis and field study, we conclude that PON topology cannot be predicted using optical power data alone when it is not a strict tree structure. For such scenarios, more related data are needed, which will be explored in our future work. 

## Figures and Tables

**Figure 1 sensors-23-03345-f001:**
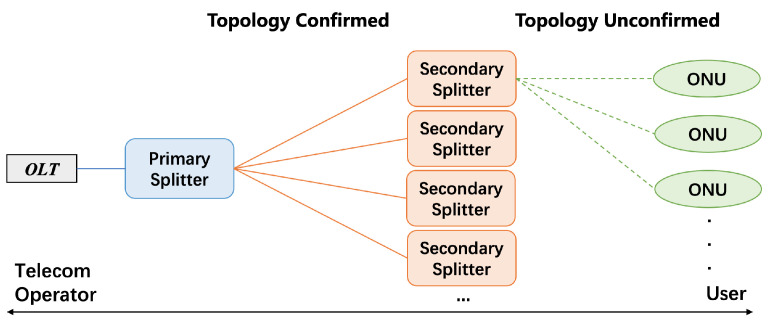
Topology of PON. OLTs are in factories of telecom operators while ONUs are in users’ rooms. The secondary splitter one ONU belongs to is registered, but not guaranteed due to passiveness and poor management. The higher level is confirmed.

**Figure 2 sensors-23-03345-f002:**
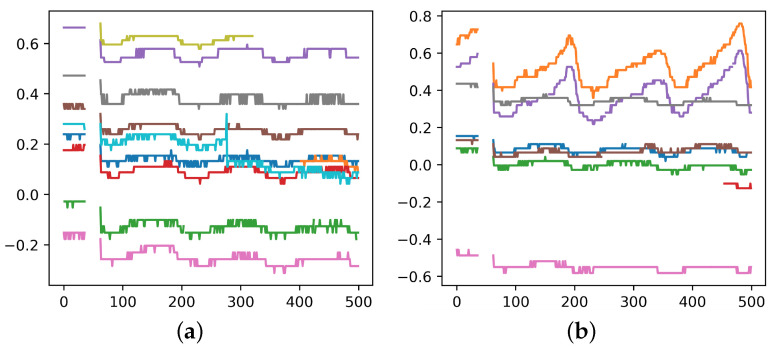
Waveforms of ONU’s optical power. The vertical axis represents the optical power. The horizontal axis represents the sampling point, which is sampled every 10 min. Each subplot shows the situation of one secondary splitter, and curves of different colors reflect the waveforms of different ONUs under the same secondary splitter. Similarity of two waveforms indicates that the two ONUs are suffering the same perturbation. (**a**) Ideal perturbation. (**b**) Non-ideal perturbation.

**Figure 3 sensors-23-03345-f003:**
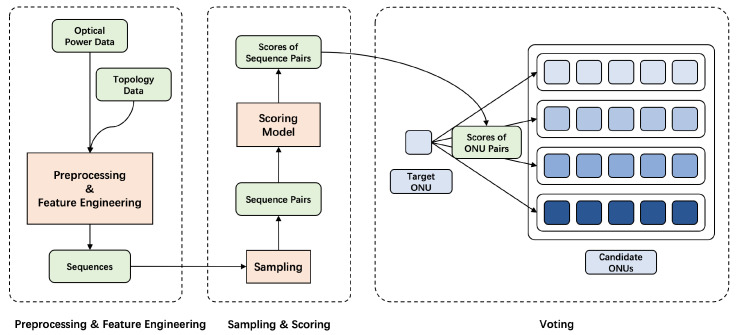
Structure of base solution. Optical power and topology data are firstly combined and preprocessed to get sequences. Then the scoring model gives the scores for the similarity of each sampled sequence pair. Using the scores, the voting algorithm determines the true label (secondary splitter) of the target ONU.

**Figure 4 sensors-23-03345-f004:**
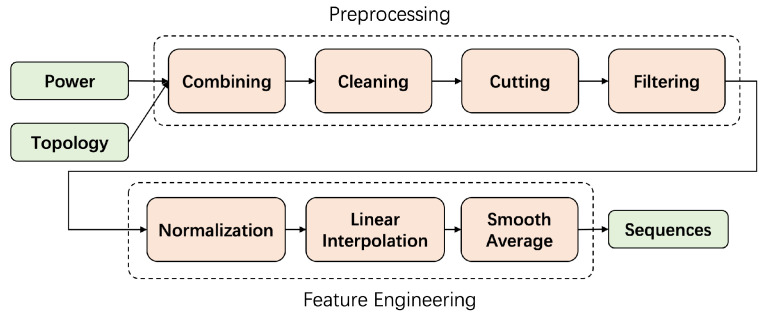
Preprocess and feature engineering.

**Figure 5 sensors-23-03345-f005:**
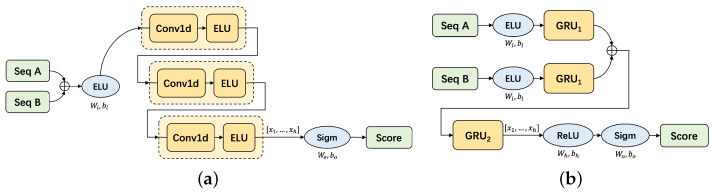
Composition of GCE-Scorer. (**a**) CNN-based scoring model. (**b**) GRU-based scoring model.

**Figure 6 sensors-23-03345-f006:**
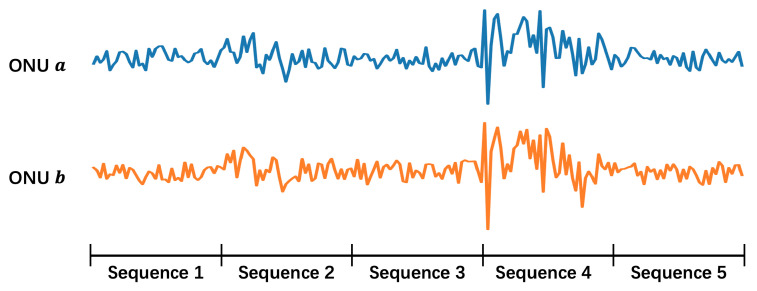
Max voting from sequences to ONUs. Perturbation occurs randomly and occasionally, so we choose the sequence pair with the most similarity as the scores of ONU pair.

**Figure 7 sensors-23-03345-f007:**
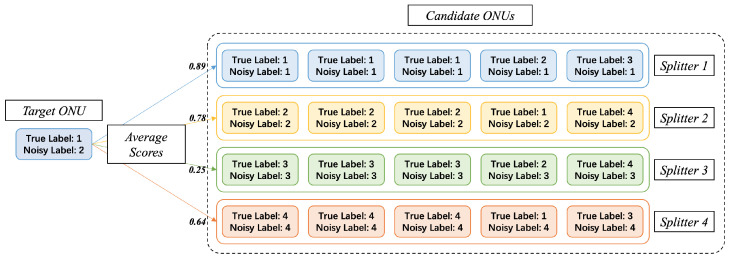
Average voting from ONUs to secondary splitters. Noisy label is provided by RMS. True label is temporarily unknown to us, but it exists objectively and is to be found.

**Figure 8 sensors-23-03345-f008:**
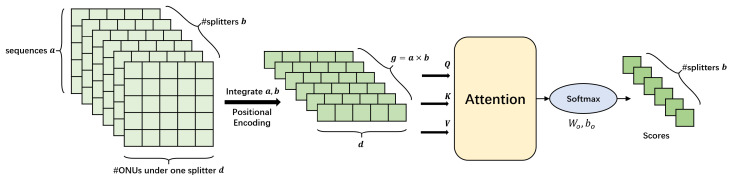
Principle of TransVoter. All scores of sequence pairs concerning one ONU are aggregated and processed to predict its label.

**Figure 9 sensors-23-03345-f009:**
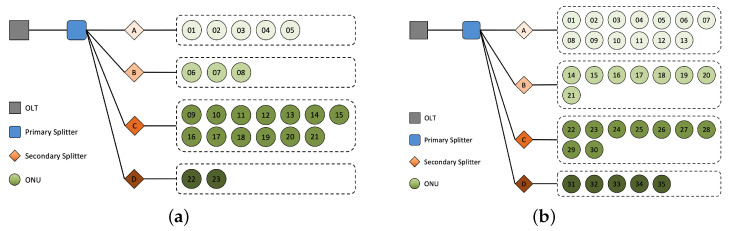
Ground truth topology of OLT 0 and OLT 1. (**a**) OLT 0. (**b**) OLT 1.

**Figure 10 sensors-23-03345-f010:**
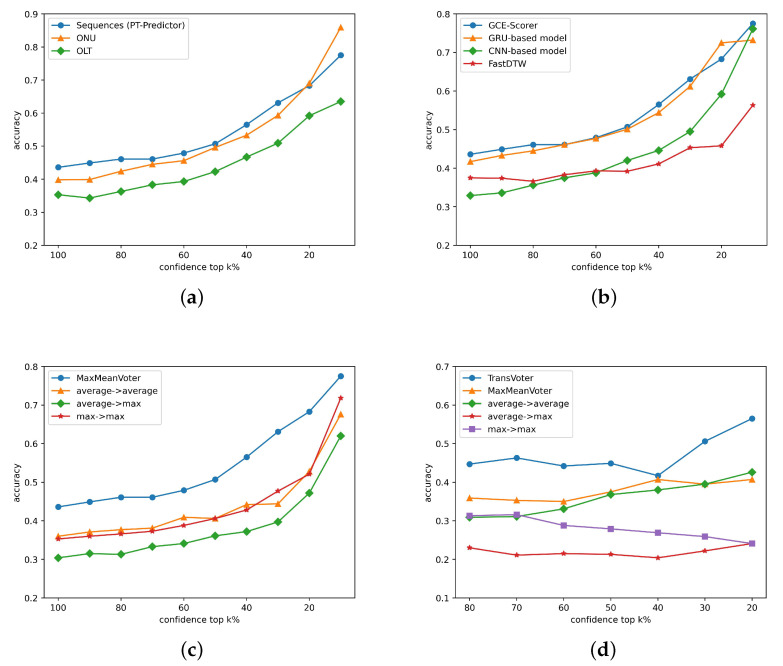
Accuracy through different methods. (**a**) Normalization levels. (**b**) Scoring methods. (**c**) Data-based voters. (**d**) Voting methods.

**Figure 11 sensors-23-03345-f011:**
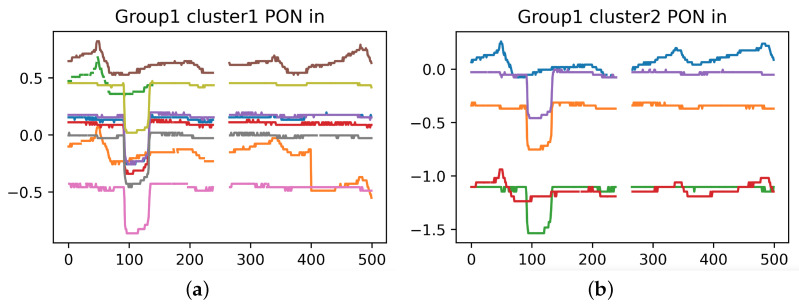
Waveforms of OLT 1. The vertical axis represents the optical power. The horizontal axis represents the sampling point, which is sampled every 10 min. Each subplot shows the situation of one secondary splitter, and curves of different colors reflect the waveforms of different ONUs under the same secondary splitter. Similarity of two waveforms indicates that the two ONUs are suffering the same perturbation. (**a**) Secondary Splitter 1. (**b**) Secondary Splitter 2.

**Table 1 sensors-23-03345-t001:** OLTs in our experiments.

OLT	#Weeks	#Class1	#Class2	#ONU	#Sequences
0	29	1	4	23	1652
1	29	1	4	35	3029
5	4	3	24	70	428
8	4	3	22	65	368
10	2	3	23	74	236
12	2	3	24	95	294
13	2	4	16	99	300

**Table 2 sensors-23-03345-t002:** Accuracy improvements on different OLTs.

	OLTs	OLT 0 (60% Coverage)	OLT 1 (60% Coverage)	OLT 5, 8, 10, 11, 12 (60% Coverage)
Methods				
Previous method		0.615	**0.524**	0.209
Base solution		0.692	0.476	0.214
PT-Predictor		**0.846**	0.476	**0.357**

**Table 3 sensors-23-03345-t003:** Accuracy through different levels of normalization.

	Coverage	100%	90%	80%	70%	60%	50%	40%	30%
Levels of Norm									
Seq (PT-Predictor)		**0.436**	**0.449**	**0.461**	**0.461**	**0.479**	**0.507**	**0.565**	**0.631**
ONU		0.398	0.399	0.424	0.445	0.456	0.496	0.533	0.593
OLT		0.353	0.343	0.363	0.383	0.393	0.423	0.467	0.509

**Table 4 sensors-23-03345-t004:** Accuracy through different scoring methods.

	Coverage	100%	90%	80%	70%	60%	50%	40%	30%
Scoring Methods									
GCE-Scorer		**0.436**	**0.449**	**0.461**	**0.461**	**0.479**	**0.507**	**0.565**	**0.631**
GRU-based model		0.417	0.433	0.445	0.461	0.477	0.501	0.544	0.612
CNN-based model		0.329	0.336	0.356	0.375	0.388	0.420	0.446	0.495
FastDTW		0.375	0.374	0.366	0.383	0.393	0.392	0.411	0.453

**Table 5 sensors-23-03345-t005:** Accuracy through different voting methods.

	Coverage	100%	90%	80%	70%	60%	50%	40%	30%
Voting Methods									
MaxMeanVoter		**0.436**	**0.449**	**0.461**	**0.461**	**0.479**	**0.507**	**0.565**	**0.631**
average→average		0.360	0.371	0.377	0.381	0.409	0.406	0.442	0.444
average→max		0.304	0.315	0.313	0.333	0.341	0.361	0.372	0.397
max→max		0.353	0.360	0.366	0.373	0.388	0.406	0.428	0.477

**Table 6 sensors-23-03345-t006:** Accuracy on re-sampled datasets through different voting methods.

	Coverage	80%	70%	60%	50%	40%	30%	20%
Voting Methods								
TransVoter		**0.447**	**0.463**	**0.442**	**0.449**	**0.417**	**0.506**	**0.565**
MaxMeanVoter		0.359	0.353	0.350	0.375	0.407	0.395	0.407
average→average		0.309	0.311	0.331	0.368	0.380	0.395	0.426
average→max		0.230	0.211	0.215	0.213	0.204	0.222	0.241
max→max		0.313	0.316	0.288	0.279	0.269	0.259	0.241

**Table 7 sensors-23-03345-t007:** Accuracy on different OLTs of different collection time.

	Time	3 w	4 w	5 w	6 w	7 w	8 w	9 w	10 w	3 m
OLTs		(100% Coverage)								
OLT 0		0.1	0.25	0.25	0.4	0.45	0.5	0.5	0.55	0.609
OLT 5		0.4	0.446	/	/	/	/	/	/	/
OLT 8		0.345	0.373	/	/	/	/	/	/	/

## Data Availability

The data presented in this study are available on request from the corresponding author. The data are not publicly available due to privacy.

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
