# Peer review of "Time-Series Representation Learning in Topology Prediction for Passive Optical Network of Telecom Operators"

_sensors, 2023, doi:10.3390/s23063345_

Round 1
Reviewer 1 Report
The manuscript is interresting and well written. Check just some gramatical
errors before the publication
Author Response
Point 1: Check just some grammatical errors before the publication.
Response 1: Thanks for your suggestions, and we have reviewed the whole paper and correct some grammatical errors, as is shown in our revised manuscript uploaded.
Reviewer 2 Report
Remarks:
1. fig. 2 and fig.10: it is necessary to give explanations for the curves
2. line. 153: Specify what type of filtration was used/filter type?. Why the interval of 10 minutes was chosen, give a ref. (option) or a description (priority).
3. eq. 4 and 5 necessary to after them specification of all symbol that used in it.
4. check conformity template in part: Supplementary Materials, Supplementary Materials, Funding, Institutional Review Board Statement, Informed Consent Statement, Data Availability Statement, Acknowledgments, Conflicts of Interest
Author Response
Point 1: fig. 2 and fig.10: it is necessary to give explanations for the curves.
Response 1: Thanks for your suggestions, and we have added more detailed explanations of the curves for both fig. 2 and fig. 10.
Point 2: line. 153: Specify what type of filtration was used/filter type?. Why the interval of 10 minutes was chosen, give a ref. (option) or a description (priority).
Response 2: We filter out the sequences with null rate higher than 20%, and we add an Appendix explaining details about it. The interval of 10 minutes is determined by the acquisition frequency of the probe in ONU, which is fixed due to the probe's physical design.
Point 3: eq. 4 and 5 necessary to after them specification of all symbol that used in it.
Response 3: We have added specification of all symbols, as is shown in the revised manuscript uploaded.
Point 4: check conformity template in part: Supplementary Materials, Supplementary Materials, Funding, Institutional Review Board Statement, Informed Consent Statement, Data Availability Statement, Acknowledgments, Conflicts of Interest.
Response 4: Thanks for your reminder, and we have checked the above items.
Reviewer 3 Report
Dear editor,
In this paper, PT-Predictor and neural networks are used to predict PON topology through representation learning on its optical power data. It is very interesting. The results showed and the conclusion drawn by the authors are reasonable and prudent. The paper is well organized. The paper is suitable published in this journal. I hope authors can continue to study and apply it to reality.
Author Response
Point 1: I hope authors can continue to study and apply it to reality.
Response 1: Thanks for your approval, and we will continue this study until it is applied to reality.